# Identification of HIV transmitting CD11c$^+$ human epidermal dendritic cells

Kirstie M. Bertram[1,2,14], Rachel A. Botting[1,2,3,14], Heeva Baharlou[1,2,14], Jake W. Rhodes [1,2], Hafsa Rana[1,2], J. Dinny Graham[1,2], Ellis Patrick [1,2], James Fletcher[3], Toby M. Plasto[1,2], Naomi R. Truong[1,2], Caroline Royle[1,2], Chloe M. Doyle [1,2], Orion Tong[1,2], Najla Nasr[1,2], Laith Barnouti[4], Mark P. Kohout[4], Andrew J. Brooks[5], Michael P. Wines[6], Peter Haertsch[7], Jake Lim[8], Martijn P. Gosselink [1,5], Grahame Ctercteko[1,5], Jacob D. Estes[9], Melissa J. Churchill[10], Paul U. Cameron [11], Eric Hunter [12], Muzlifah A. Haniffa [3,13], Anthony L. Cunningham[1,2] & Andrew N. Harman [1,2]

Langerhans cells (LC) are thought to be the only mononuclear phagocyte population in the epidermis where they detect pathogens. Here, we show that CD11c$^+$ dendritic cells (DCs) are also present. These cells are transcriptionally similar to dermal cDC2 but are more efficient antigen-presenting cells. Compared to LCs, epidermal CD11c$^+$ DCs are enriched in anogenital tissues where they preferentially interact with HIV, express the higher levels of HIV entry receptor CCR5, support the higher levels of HIV uptake and replication and are more efficient at transmitting the virus to CD4 T cells. Importantly, these findings are observed using both a lab-adapted and transmitted/founder strain of HIV. We also describe a CD33$^{low}$ cell population, which is transcriptionally similar to LCs but does not appear to function as antigen-presenting cells or acts as HIV target cells. Our findings reveal that epidermal DCs in anogenital tissues potentially play a key role in sexual transmission of HIV.

[1] Centre for Virus Research, The Westmead Institute for Medical Research, 176 Hawkesbury Road, Westmead, New South Wales 2145, Australia. [2] The University of Sydney, Sydney 2006 New South Wales, Australia. [3] Institute of Cellular Medicine, Newcastle University, Newcastle upon Tyne NE2 4HH, UK. [4] Australia Plastic Surgery, 185-211, Broadway, Sydney, New South Wales 2007, Australia. [5] Westmead Hospital, Westmead, New South Wales 2145, Australia. [6] Royal North Shore Hospital, Reserve Rd, St Leonards, New South Wales 2065, Australia. [7] Burns Unit, Concord Repatriation General Hospital, Sydney 2139 New South Wales, Australia. [8] Dr Jake Lim PLC, Shop 12, Cnr of Aird & Marsden Street, Parramatta, New South Wales 2150, Australia. [9] AIDS and Cancer Virus Program, Frederick National Laboratory for Cancer Research, Leidos Biomedical Research Inc, Frederick, MD 21702, USA. [10] School of Health and Biomedical Sciences, College of Science, Engineering and Health, RMIT University, Melbourne, Victoria 3001, Australia. [11] The Peter Doherty Institute for Infection and Immunity, 792 Elizabeth Street, Melbourne, Victoria 3000, Australia. [12] Emory Vaccine Center, 954 Gatewood Road, Atlanta, GA 30329, USA. [13] Department of Dermatology, Royal Victoria Infirmary, Newcastle Hospitals NHS Foundation Trust, Newcastle upon Tyne NE2 4LP, UK. [14] These authors contributed equally: Kirstie M. Bertram, Rachel A. Botting, Heeva Baharlou. Correspondence and requests for materials should be addressed to A.N.H. (email: andrew.harman@sydney.edu.au)

Mononuclear phagocytes (MNP) are antigen-presenting cells (APCs) and represent the first line of contact between the immune system and invading pathogens in the tissues that form the portals of pathogen entry. In the case of sexually transmitted pathogens, such as HIV, these are the genital and anorectal (anogenital) tracts which are made up of various tissues comprised of either skin or mucosal tissue. Skin (outer foreskin, glans penis and labia) and type II mucosal tissues (inner foreskin, vagina, ectocervix, fossa navicularis and anal canal) both contain a stratified squamous epithelium predominantly made up of keratinocytes, which forms a formidable physical barrier against invading pathogens. However, HIV penetration and infection across this surface has been demonstrated in human ex vivo genital tissue explants from the vagina, ectocervix and foreskin[1–6].

Although several subsets of MNPs are known to exist in the dermis and lamina propria[7], only a single subset has been described in healthy epidermis called Langerhans cells (LC). LCs form an interconnected network and are known to bind HIV via the C-type lectin receptor (CLR) langerin[8,9]. LCs are reported to be involved in sexual transmission of HIV but their role needs to be more clearly defined[1–3,8–10].

As APCs, LCs express high levels of CD45 and HLA-DR on their surface as well as CD1a[3,9] and the HIV binding CLR langerin[2]. They express low levels of the integrin CD11c, which is highly expressed by murine conventional DCs and human cDC2, and do not express the HIV binding CLRs Dendritic Cell-Specific Intercellular adhesion molecule-3-Grabbing Nonintegrin (DC-SIGN) or mannose receptor (MR). These cells are believed to be the only MNP found within healthy epidermis. In inflamed epidermis an additional MNP subset has been described referred to as Inflammatory Dendritic Epidermal Cell (IDEC), which can be discerned from LCs by their high expression of CD11c and MR[11–14].

Recently we found that trypsin, which is the most common enzyme used to isolate LCs from the epidermis, cleaved CD11c as well as multiple pathogen binding receptors and the HIV entry receptor CD4[15]. This may have affected identification of epidermal MNPs and their HIV transmission capacity. In this study, using our optimised MNP isolation protocols, we identify a subset of MNPs in healthy epidermis, in addition to LCs. These cells express high CD11c and MR which we refer to as epidermal CD11c+ DCs. Importantly, they are enriched in anogenital epidermis where they interact with HIV preferentially to LCs. These cells also express higher amounts of CCR5 on their surface compared to LCs, are more permissive for infection with HIV and more efficient at transferring HIV to CD4 T cells. We also identify a population of cells that are phenotypically similar to LCs but express lower levels of CD45, HLA-DR and CD33. However, they do not stimulate T cell proliferation or act as HIV target cells. We refer to these cells as epidermal CD33low cells.

## Results

### Three subsets of mononuclear phagocytes in healthy epidermis.
Previously we have optimised skin enzymatic digestion protocols to liberate MNPs with an immature phenotype and minimal surface receptor cleavage. Critically, we found that trypsin cleaved many MNP defining markers including CD1c and CD11c whereas collagenase did not[15]. Using this method we digested fresh healthy abdominal epidermal sheets with collagenase and identified three CD45+/HLA-DR+/CD3−/CD19− cell populations using flow cytometry (Fig. 1a). LCs could be discerned by their high surface expression of CD1a, langerin and CD33, low CD11c expression and absence of MR expression. A second CD1a+langerin+CD11clowMR− cell type could be distinguished

from LCs by their much lower surface expression of CD33 as well as lower levels of CD45, HLA-DR and CD1a. We refer to these cells as epidermal CD33low cells. Finally, we observed a CD11chigh cell that also expressed MR, higher levels of HLA-DR and CD1c and slightly lower levels of CD1a than LCs and in most cases did not express langerin. We refer to these as epidermal CD11c+ DCs. All three epidermal populations did not express CD14 or CLEC9A demonstrating that they were not contaminating dermal cDC1 or monocyte-derived macrophages. We then investigated how the epidermal MNP subsets related to dermal MNPs using our gating strategy that identifies all known human dermal MNP subsets[15] (Supplementary Fig. 1a). Interestingly, we found that epidermal CD11c+ DCs aligned closely to dermal cDC2. We then compared trypsin and collagenase digestion and confirmed that CD11c+ cells were not detected when using trypsin, which cleaves CD11c, to digest epidermis (Fig. 1b).

We next used flow cytometry to determine the relative abundance of each cell type within the CD45+HLA-DR+ population in abdominal epidermis using 75 healthy donors (Fig. 1c). Although there was some donor variability, LCs were the most abundant cell type followed by epidermal CD11c+ DCs and then epidermal CD33low cells.

We next used RNAseq to compare the transcriptional profiles of epidermal CD11c+ DCs and CD33low cells to all known subsets of human skin MNPs (Fig. 2). Consistent with the flow cytometry profiling (Supplementary Fig. 1a) we found that epidermal CD11c+ DCs were transcriptionally very similar to both known subsets of dermal cDC2 and could not be distinguished by principle component analysis. In contrast, CD33low cells were transcriptionally very similar to LCs (Fig. 2a). Heatmaps constructed using the top 40 differential genes between subsets showed similar clustering (Fig. 2b), which was confirmed by QPCR (Supplementary Fig. 2).

We next assessed the presence of epidermal MNPs in situ. Firstly, we carried out intracellular staining and assessed the effect that this had on identifying each population by flow cytometry (Supplementary Fig. 1b). Although, we could still clearly discern a discrete CD11c+ cell population, we were unable to visualise a CD33low population as all HLA-DR+CD1a+ epidermal cells expressed CD33 at equal levels intracellularly. We next stained abdominal tissue blocks with CD1c, CD11c, MR and langerin and confirmed using immunofluorescence microscopy that in contrast to LCs, epidermal CD11c+ DCs were bright for CD11c and CD1c, expressed low levels of MR and did not express langerin (Fig. 3a). Although both populations could be clearly visualised, epidermal CD11c+ DCs were almost always observed close to the basement membrane whereas LCs were dispersed throughout the epithelium (Fig. 3b). Furthermore, we observed that epidermal CD11c+ cells clustered in distinct regions, which were concomitant with a decrease in LC density.

Finally, we FACS-sorted each epidermal MNP population, as well as dermal CD11c+ cDC2, and examined their morphology using Giemsa staining (Fig. 3c). Each epidermal population was distinct but possessed morphological characteristics of DCs and not lymphocytes, macrophages or mast cells[16]. LCs could be discerned by their dendrite structures; epidermal CD33low cells were the smallest subset and epidermal CD11c+ DCs were similar to dermal CD11c+ cDC2.

### Epidermal mononuclear phagocyte pathogen receptor expression.
A defining feature of MNPs is their ability to capture pathogens via an array of lectin receptors (CLRs and Siglecs) expressed on their surface[17,18]. Using the appropriate blend of collagenase to liberate cells from healthy abdominal tissue[15] we

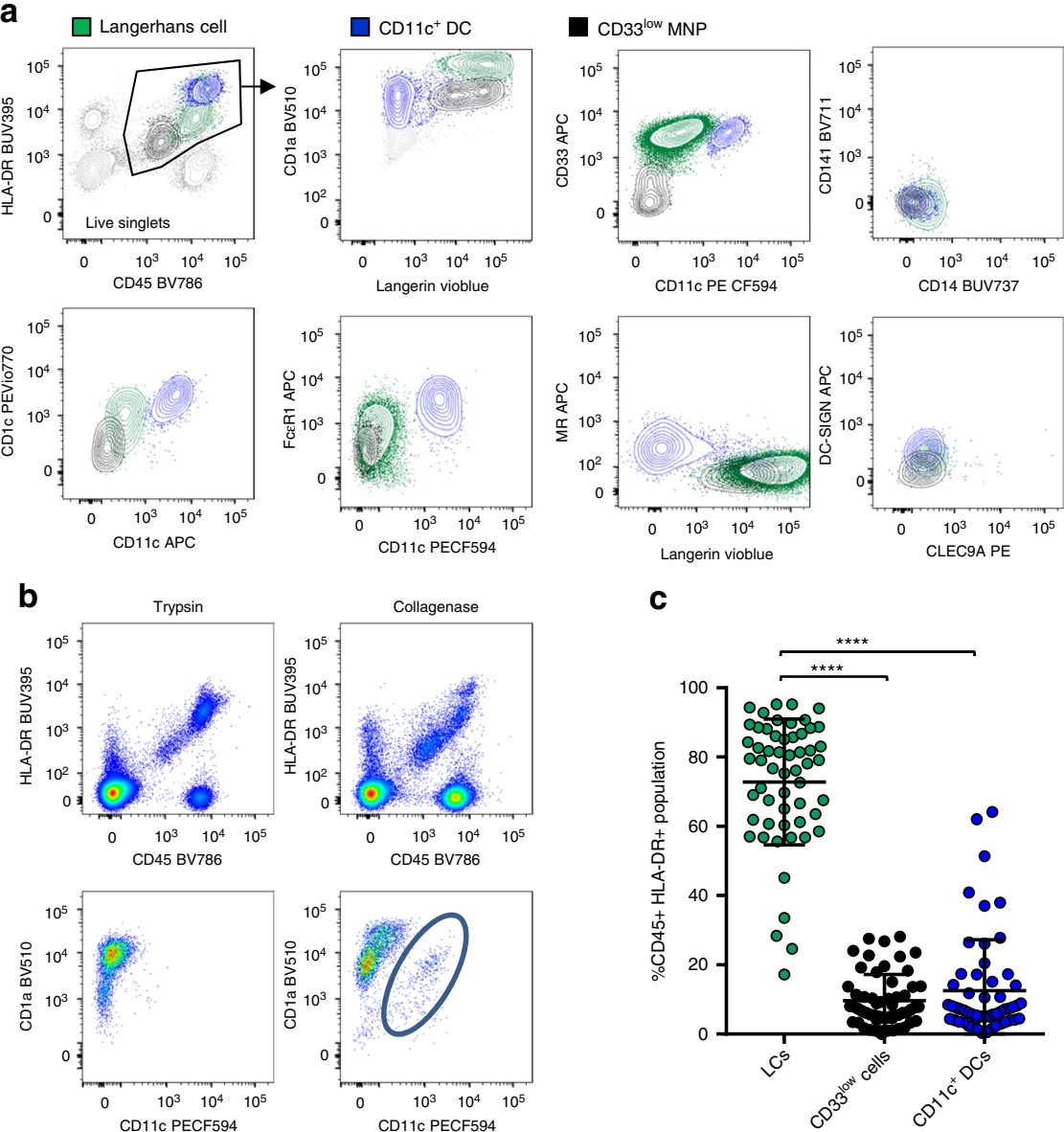

**Fig. 1** Definition of distinct human epidermal mononuclear phagocytes by flow cytometry. Mononuclear phagocytes were liberated from human epidermis using trypsin or Type IV collagenase. **a** Flow cytometry was used to define three subsets of CD3−/CD19epidermal MNPs from collagenase digested tissue (i) Langerhans cells (green) were defined as HLA-DR[intermediate], CD1a[very high], langerin[high], CD33+, CD11c−, MR− (ii) epidermal CD11c+ DCs (blue) were defined as HLA-DR[high], CD1a+, langerin[low], CD11c+, MR+ (iii) epidermal CD33[low] cells (black) were defined as HLA-DR[low], CD1a+, langerin[high], CD33[low], CD11c−, MR−. Representative data of 75 individual donors is shown. **b** CD11c expression was compared on cells derived from trypsin and collagenase digestion. The blue circle indicates the CD11c+ cells detected in collagenase digested tissue that are not observed in trypsin digested tissue. Representative data of five individual donors is shown. **c** The relative proportion of each subset CD45+ HLA-DR+ gate after collagenase digestion was determined and is shown as a mean ± the standard deviation with each dot representing an individual donor. Statistics were generated using the Mann–Whitney test. *$p <$ 0.05; **$p < 0.01$; ***$p < 0.001$; ****$p < 0.0001$ ($n = 75$)

next defined the lectin surface expression profiles of each epidermal MNP subset using flow cytometry and also compared them to dermal cDC2. Furthermore, as LCs have been shown to be infected by HIV, we also measured the surface expression of the HIV entry receptors CD4, CCR5 and CXCR4 (Table 1 and Supplementary Fig. 3). LCs expressed high levels of langerin, CLEC5A and DCIR, moderate levels of DEC205, Siglec-3 (CD33) and Siglec-9, low levels of CD4 and CCR5, and did not express MR, DC-SIGN or CXCR4. Other than expressing lower levels of Siglec-3 (CD33), epidermal CD33low cells showed an almost identical pattern of lectin receptor expression to LCs. However they did not express any of the HIV entry receptors CD4, CCR5

or CXCR4. Epidermal CD11c+ DCs expressed all the lectins expressed by LCs except for langerin. In addition, and distinct from LCs, CD11c+ DCs expressed CLEC4M (L-SIGN), CLEC10A, CLEC12A and MR as well as Siglec-5 and much higher levels of Siglec-9. Consistent with their transcriptional profiles (Fig. 2) this was a similar pattern of surface expression to dermal cDC2 but the two populations did differ in that dermal cDC2 expressed low levels of CLEC8A, Siglec-1, Siglec-6 and Siglec-16 and much higher levels of MR. Importantly, we also observed that epidermal CD11c+ DCs expressed much higher levels of the HIV entry receptor CCR5 than any other MNP (including dermal cDC2).

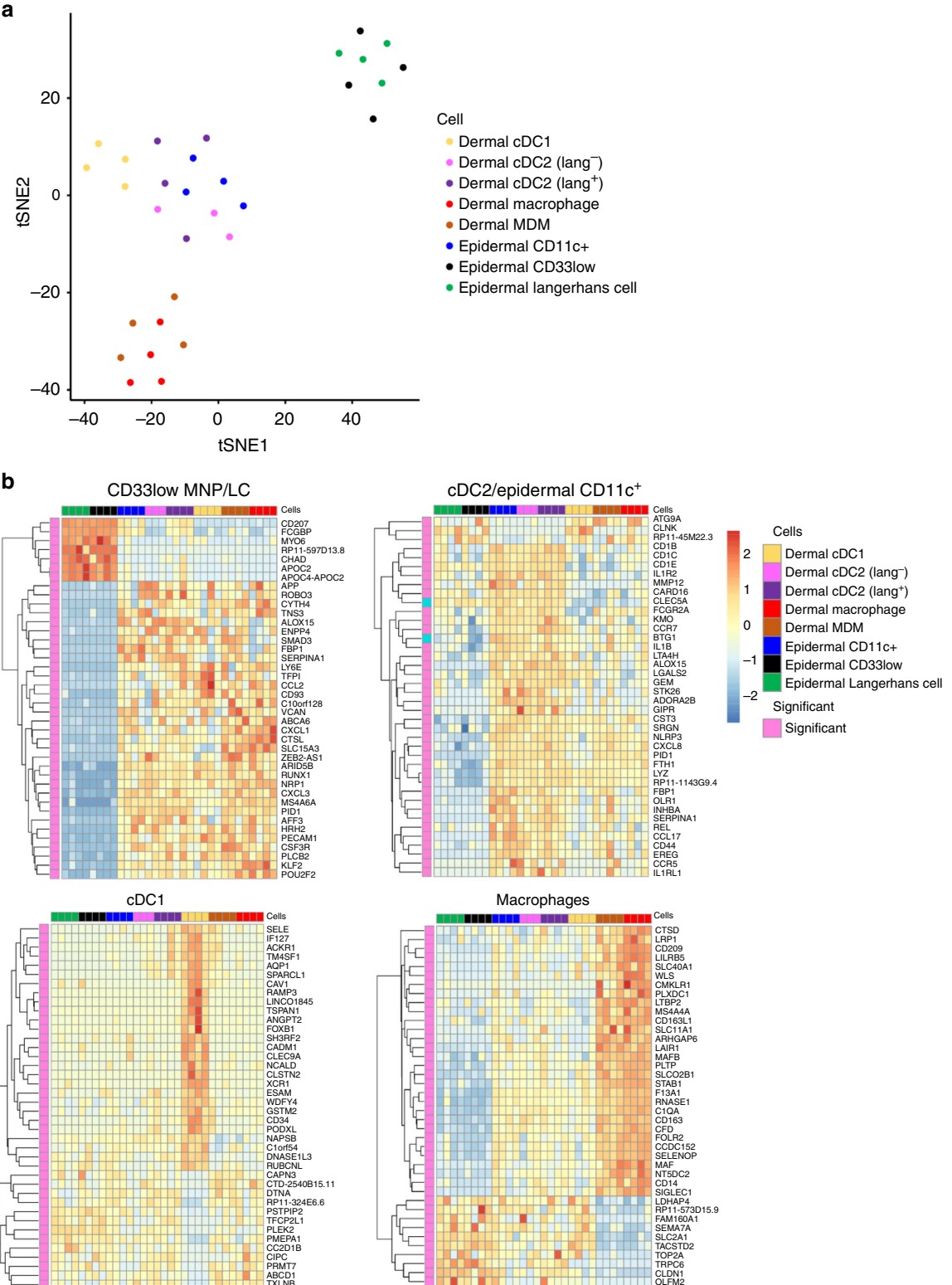

**Fig. 2** Transcriptional profiling of epidermal and dermal human mononuclear phagocytes. Epidermal LCs, CD11c[+]DCs and CD33[low] cells and dermal cDC1, cDC2 (langerin[+] and [−]), monocyte derived macrophages and tissue resident macrophages were FACS sorted from 4 individual donors. RNA was amplified and RNAseq carried out. **a** A tSNE plot highlights similarities between the cell subsets. **b** Moderated linear models were used to identify genes whose high or low expression characterised each subset. Heat maps display expression values of the top 40 genes that uniquely identify; CD33low MNPs and LCs; cDC2 and epidermal CD11c; cDC1; and monocyte-derived macrophages and tissue resident Macrophages

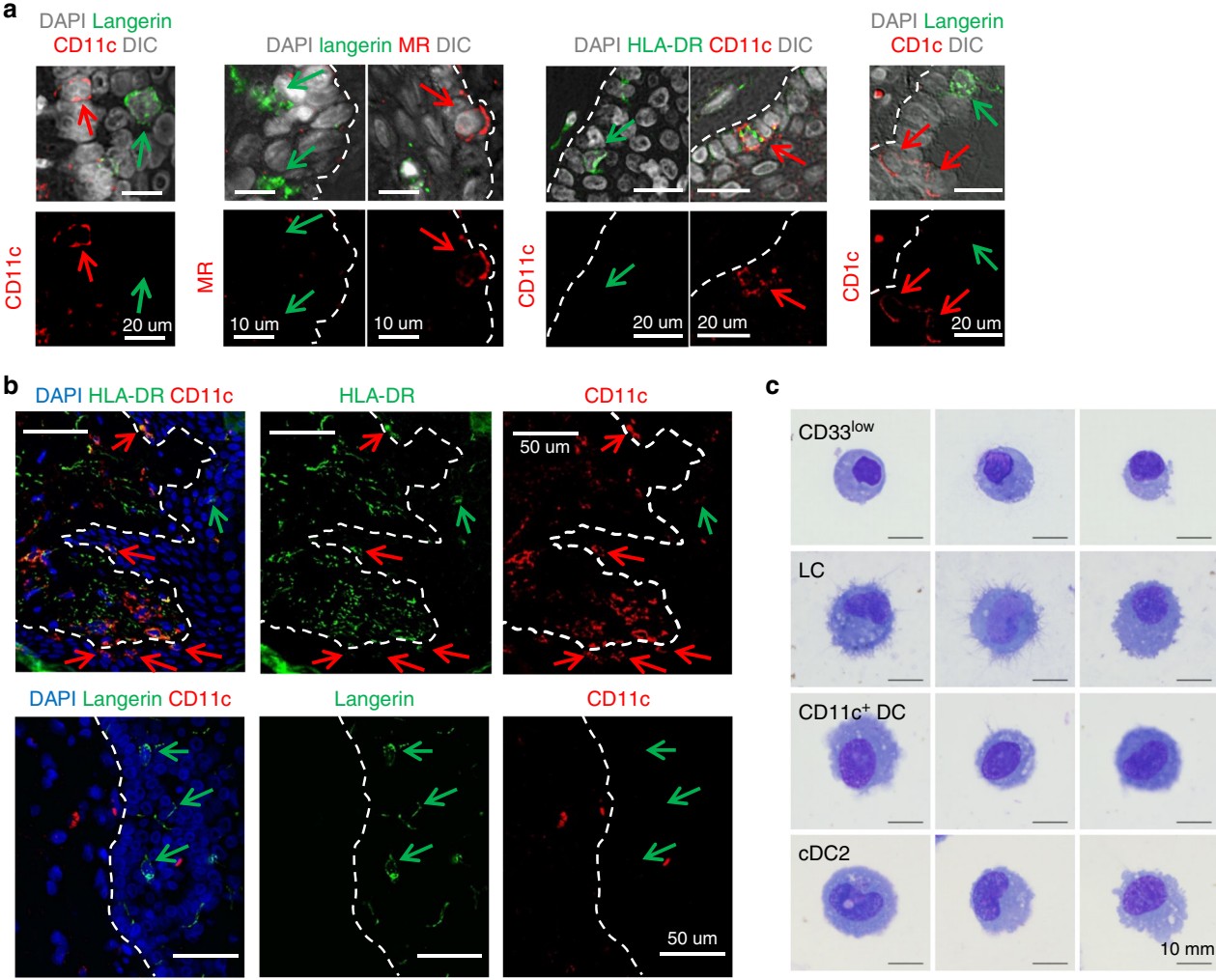

**Fig. 3** Visualisation of CD11c+ dendritic cells in situ and morphological staining of epidermal MNPs. **a**, **b** Human abdominal tissue was sectioned and stained for HLA-DR, CD11c, CD1c, MR and langerin for fluorescence microscopy. All images representative of at least 3 different donors. **a** Epidermal CD11c+ DCs could be discerned from LCs by their expression of MR, brighter expression of CD1c and CD11c and lack of langerin expression. **b** Epidermal CD11c+ DCs (white arrows) could be discerned from LCs (red arrows) in situ as HLA-DR+, CD11cbright cells and were almost always situated close to the basement membrane. **c** Epidermal sheets were digested with collagenase Type IV and each epidermal HLA-DR+CD45+ MNP population was FACS sorted and a Giemsa stain carried out for morphological examination

**Mononuclear phagocyte functional characterisation**. In response to pathogen detection MNPs undergo a process of maturation whereby they secrete a specific range of cytokines and upregulate molecules involved T cell binding (CD54) and costimulatory molecules (CD80, CD83 and CD86) which allows them to activate T cells to undergo proliferation[7]. We therefore used a cytometric bead array assay to define the cytokines that each epidermal subset and dermal cDC2 secreted in response to a TLR agonist cocktail (Fig. 4a). We also measured CD54 and co-stimulatory molecule surface expression on each epidermal subset (Fig. 4b) and carried out T cell alloproliferation assays (Fig. 4c). We found that each subset had a unique cytokine expression profile. LCs produced IL-8 and low levels of IL-1$_\alpha$ and TNF whereas epidermal CD33low cells did not secrete any cytokines to detectable levels. Epidermal CD11c+ DCs and dermal DC2 showed a similar cytokine expression profile and produced IL-1$_\beta$, IL-6, IL-8, IL-10 and TNF. However, notably, epidermal CD11c+ DCs secreted higher levels in all cases which reached statistical significance for IL-6, IL-8 and IL-10. cDC2 but not epidermal CD11c+ DCs produced IL-1. Consistent with their lack

of cytokine secretion we also found that epidermal CD33low cells expressed lower levels of co-stimulatory molecules on their surface and were poor inducers of naïve T cell proliferation. Epidermal CD11c+ DCs and LCs expressed CD83 and CD86 at similar levels and were efficient at inducing T cell proliferation. However, epidermal CD11c+ DCs and dermal cDC2 expressed significantly higher levels of CD80 compared to LCs and epidermal CD11c+ DCs expressed significantly higher levels CD54 then all other cell types.

**Enrichment of CD11c DCs in anogenital tissues and langerin**. As epidermal CD11c+ DCs derived from abdominal skin expressed higher levels of the HIV entry receptor CCR5 (Table 1 and Supplementary Fig. 3), we hypothesised that they would be preferential targets for HIV infection compared to the other two epidermal MNP subtypes. We therefore obtained human ano-genital tissues containing a stratified squamous epithelium that HIV may encounter during sexual transmission (labia, vagina, foreskin, glans penis, fossa navicularis, anal canal and perineum)

**Table 1 Pathogen binding receptor surface expression profiles of epidermal MNPs**

| Marker | Collagenase blend | Epidermal LCs | Epidermal CD33$^{low}$ cell | Epidermal CD11c$^+$ DC | Dermal cDC2 |
|---|---|---|---|---|---|
| CD4 | Type IV | + | − | ++ | ++ |
| CCR5 | Blend F | + | − | +++ | + |
| CXCR4 | Type IV | − | − | − | − |
| Clec4A (DCIR) | Blend F | ++++ | +++ | +++++ | +++++ |
| ClecC4D | Blend F | − | − | − | − |
| Clec4E | Blend F | + | ++ | +++ | +++ |
| Clec4K (langerin) | Blend F | +++++ | +++++ | − | − |
| Clec4L (DC-SIGN) | Blend F | − | − | − | − |
| Clec4M (L-SIGN) | Blend F | − | − | ++ | ++ |
| Clec5A | Blend F | +++++ | +++ | +++++ | +++++ |
| Clec5B | Blend F | − | − | − | − |
| Clec5C | Blend F | − | − | − | − |
| Clec6A (Dectin2) | Blend F | − | − | − | − |
| Clec8A | Blend F | − | − | − | + |
| Clec8A | Blend F | − | − | − | − |
| Clec10A | Blend F | − | − | ++ | ++ |
| Clec12A | Blend F | − | − | +++ | +++ |
| Clec13B (DEC205) | Blend F | ++ | + | + | + |
| Clec13D (MR) | Blend F | − | − | ++ | ++++ |
| Clec14A | Blend F | − | − | − | − |
| Siglec-1 | Blend F | − | − | − | + |
| Siglec-3 (CD33) | Blend F | +++++ | + | +++++ | +++ |
| Siglec-5 | Blend F | − | − | + | + |
| Siglec-6 | Blend F | − | − | − | + |
| Siglec-9 | Blend F | ++ | + | ++++ | ++++ |
| Siglec-16 | Blend F | − | − | − | + |

Ex vivo epidermal MNPs and dermal cDC2 were isolated from human skin using either Blend F or Type IV collagenase and the surface expression was determined for a range of pathogen binding receptors and HIV entry receptors by flow cytometry. The table shows the mean surface expression (MFI) of each surface marker from five independent donors. −MFI < 100, +MFI 100 − 499, ++MFI 500 − 999, +++MFI 1000 − 4999, ++++5000 − 9999, +++++MFI ≥ 10,000. Representative histograms for each marker on each cell type are shown in Supplementary Fig. 2.

and compared the relative proportions of respective epidermal MNP subsets in each tissue by flow cytometry (Fig. 5a). We found that epidermal CD11c$^+$ DCs were enriched in anogenital tissues compared to abdomen. We then confirmed our ex vivo data by showing that the relative proportions of LCs and epidermal CD11c$^+$ DCs were the same in situ in inner foreskin by florescence microscopy (Fig. 5b). Partly, to exclude the possibility that the epidermal CD11c$^+$ DCs were contaminating lamina propria CD11c$^+$ DCs, we stained for CD1a on CD11c$^+$ cells derived from both tissues as, unlike dermis, CD1a is not expressed in the lamina propria (Supplemental Fig. 1c). In contrast to lamina propria at least 50% of vaginal epithelial CD11c$^+$ cells expressed CD1a. Relevant to HIV transmission, we confirmed that epidermal CD11c$^+$ DCs expressed higher amounts of CCR5 in these tissues compared to LCs and CD33$^{low}$ cells (Fig. 5c). Finally, in contrast to abdominal tissue, between 50 and 80% of anogenital CD11c$^+$ DCs expressed langerin on their surface (Fig. 5d).

**Epidermal CD11c DCs preferentially interact with HIV in situ**. We next determined if epidermal CD11c$^+$ DCs interacted with HIV in situ. To this end we topically applied the lab-adapted HIV-1 BaL strain as well as the transmitted/founder strain HIV-1 Z3678M[19] to fresh inner foreskin explants for 2 h and used RNAscope in situ hybridization technology to investigate if epidermal CD11c$^+$ DCs interacted with the virus. The technology uses probes and a detection system that can be developed with fluorescent chromogens and can detect HIV with single virion sensitivity. This method overcomes the high background staining observed with antibodies to detect HIV and lower infectivity when using fluorescently labelled virions[20]. Unlike abdominal tissue where almost all epidermal CD11c$^+$ DCs were present in the basal layer close to the basement membrane (Fig. 4b), in inner foreskin CD11c$^+$ DCs could be visualised throughout the epithelial layer (Fig. 6a) similar to LCs (Fig. 6b). In tissue infected

with both virus strains, epidermal CD11c$^+$ DCs were visualised appearing to contain virion RNA within them (Fig. 6c) and could sometimes be seen close to the epithelial surface appearing to extend out dendritic processes to contact virions (Fig. 6c). Additional images of CD11c$^+$ DCs interacting with HIV-1$_{BaL}$ and HIV1$_{Z3678B}$ in three additional independent donors are shown in Supplemental Fig. 4a, b. Consistent with the published literature[2,3] we also observed some LCs containing HIV (Fig. 6d), which are reported to rapidly migrate out of the epidermis after infection. Finally, we quantified the relative amount of HIV that could be detected localising to both LCs and epidermal CD11c$^+$ DCs in situ using transmitted/founder strain Z3678M. We found that there was significantly more HIV-1 associated with epidermal CD11c$^+$ DCs compared to LCs (Fig. 6e). As controls, RNAscope was performed on mock tissue using HIV specific probes (Supplemental Fig. 4c), and HIV treated tissue using the bacterial dapB specific probe as a negative probe (Supplemental Fig. 4d). In both cases the incidence of false positive signals was negligible.

**HIV infection of CD11c DCs and transfer to CD4 T cells**. As we observed that epidermal CD11c$^+$ DCs appeared to take up HIV in situ (Fig. 6d) and that they expressed high CCR5 levels (Fig. 5b), we next assessed their ability to take up and become infected by HIV-1 and then transfer the virus to CD4 T cells ex vivo. To this end we sorted epidermal CD11c$^+$ DCs, CD33$^{low}$ cells and LCs as well as dermal cDC2 from abdominal skin and then infected them with the lab-adapted HIV-1 BaL strain and/or the transmitted/founder strain HIV-1 Z3678M[19].

Firstly we used flow cytometry to directly measure the amount of HIV RNA (PrimeFlow) and p24 protein contained within epidermal CD11c$^+$ DCs and LCs after 2 and 96 h of HIV exposure using HIV-1$_{Z3678M}$ (Fig. 7a). We found that epidermal CD11c$^+$ DCs contained higher levels of HIV RNA and p24 than

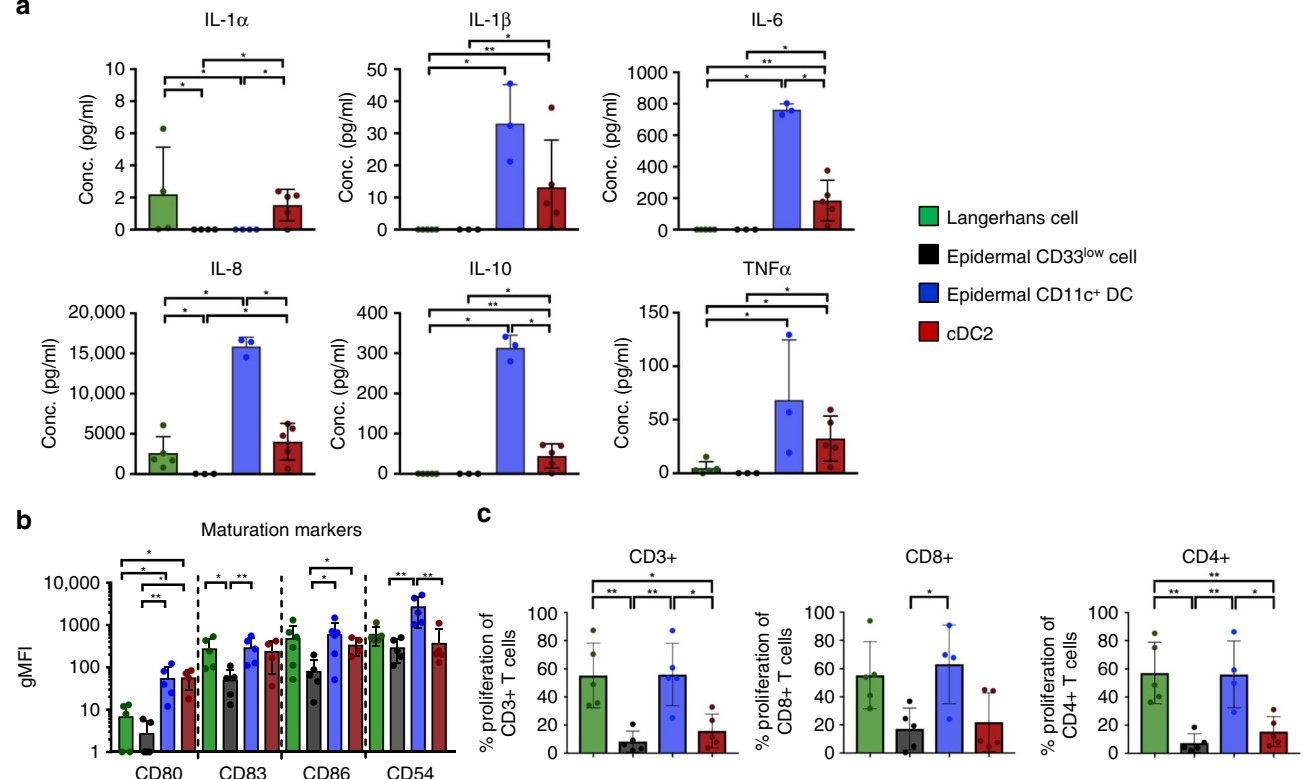

**Fig. 4** Functional phenotyping of human epidermal mononuclear phagocytes. Mononuclear phagocytes were liberated from human epidermis using Type IV collagenase. cDC2 were also isolated from dermis. **a** Each subset was FACS sorted and then cultured overnight in a cocktail of TLR agonists. The supernatants were collected and a cytometric bead array was carried out to determine the amounts of IL-1$_\alpha$, IL-1$_\beta$, IL-6, IL-8, IL-10 and TNF$_\alpha$. Results expressed as mean ± standard deviation with each dot representing an individual donor. Statistical analysis was generated using the Mann–Whitney test ($n \geq 3$). **b** The surface expression of CD54, CD80, CD83 and CD86 were determined on each cell type immediately after tissue extraction. The gMFI of each subset was calculated and plotted as the mean ± standard deviation, with each dot representing an individual donor. Statistical analysis was generated using the Mann–Whitney test ($n = 5$). **c** FACS sorted epidermal MNP subsets were cultured for 7 days at 37 °C with Celltrace Violet stained CD3$^+$ T cells at a ratio of 1:10 and analysed by flow cytometry. The results expressed as mean ± standard deviation where each dot represents an individual donor ($n = 5$). Statistical analysis was performed using the Mann–Whitney test. *$p < 0.05$; **$p < 0.01$; ***$p < 0.001$; ****$p < 0.0001$

LCs after 2 h indicating that they are more efficient at HIV uptake. We also found that they contained higher levels of p24 than LCs at 96 h indicating that they support higher levels of productive infection.

To confirm that these cells were productively infected and secreted infectious virus we collected supernatants from cells infected with both HIV-1$_{BaL}$ and HIV-1$_{Z3678M}$ for 96 h and cultured them with TZMBL1 CD4 T cells and measured their infectivity (Fig. 7b). As expected TZMBL1 CD4 T cells cultured with HIV-1 BaL infected cell supernatants derived from epidermal CD11c$^+$ DCs became significantly more infected than those cultured with supernatants derived from LCs, CD33$^{low}$ cells and cDC2. Similarly, TZMBL1 CD4 T cells cultured with HIV-1 Z3678M derived from epidermal CD11c$^+$ DCs became significantly more infected than those cultured with supernatants derived from LCs.

Next, we measured the ability of each cell type to transfer the virus to CD4 T cells using our MNP-T cell transfer assay[9,21–23] and optimised MNP culture conditions[15] (Fig. 8a). Consistent with their lack of CD4 and CCR5 expression and low levels of infectivity, CD33$^{low}$ cells were unable to transfer the virus to CD4 T cells. Consistent with their enhanced ability to take up and become infected with HIV$_{BaL}$, epidermal CD11c$^+$ DCs were the most efficient cells at transferring the virus to CD4 T cells after 96 h. Similar results were observed using HIV$_{Z3678M}$. In order to rule out the unlikely possibility that transfer of HIV from

epidermal CD11c$^+$ DCs to CD4 cells was a result of virus retained on the cell surface, we used our two phase transfer assay and showed that transfer of HIV$_{BaL}$ to CD4 T cells by epidermal CD11c$^+$ DCs declined rapidly with time between 2 and 24 h before a second phase transfer began to occur after 72 h as a result of de novo produced virions (Fig. 8b). Furthermore, to show that transfer of HIV-1 to CD4 T cells is dependent on them becoming productively infected via gp120/CCR5 mediated entry, we pre-treated epidermal CD11c$^+$ DCs and LCs with the CCR5 blocking drug maraviroc and showed that this completely blocked their ability to mediate transfer of HIV$_{BaL}$ to CD4 T cells after 96 h (Fig. 8c).

## Discussion
We report here the observation of three human CD45$^+$ HLA-DR$^+$ epidermal subsets in human abdominal and anogenital epidermis, two of which have not been previously reported in healthy tissue to the best of our knowledge. We refer to these subsets as LCs, epidermal CD11c$^+$ DCs and epidermal CD33$^{low}$ cells. We transcriptionally profiled each cell type and then functionally characterised them in terms of their pathogen binding receptor expression, cytokine secretion profiles, costimulatory marker expression and ability to induce T cell proliferation. Critically, we showed that epidermal CD11c$^+$ DCs are able to interact with both lab-adapted and transmitted/founder HIV strains at the anogenital

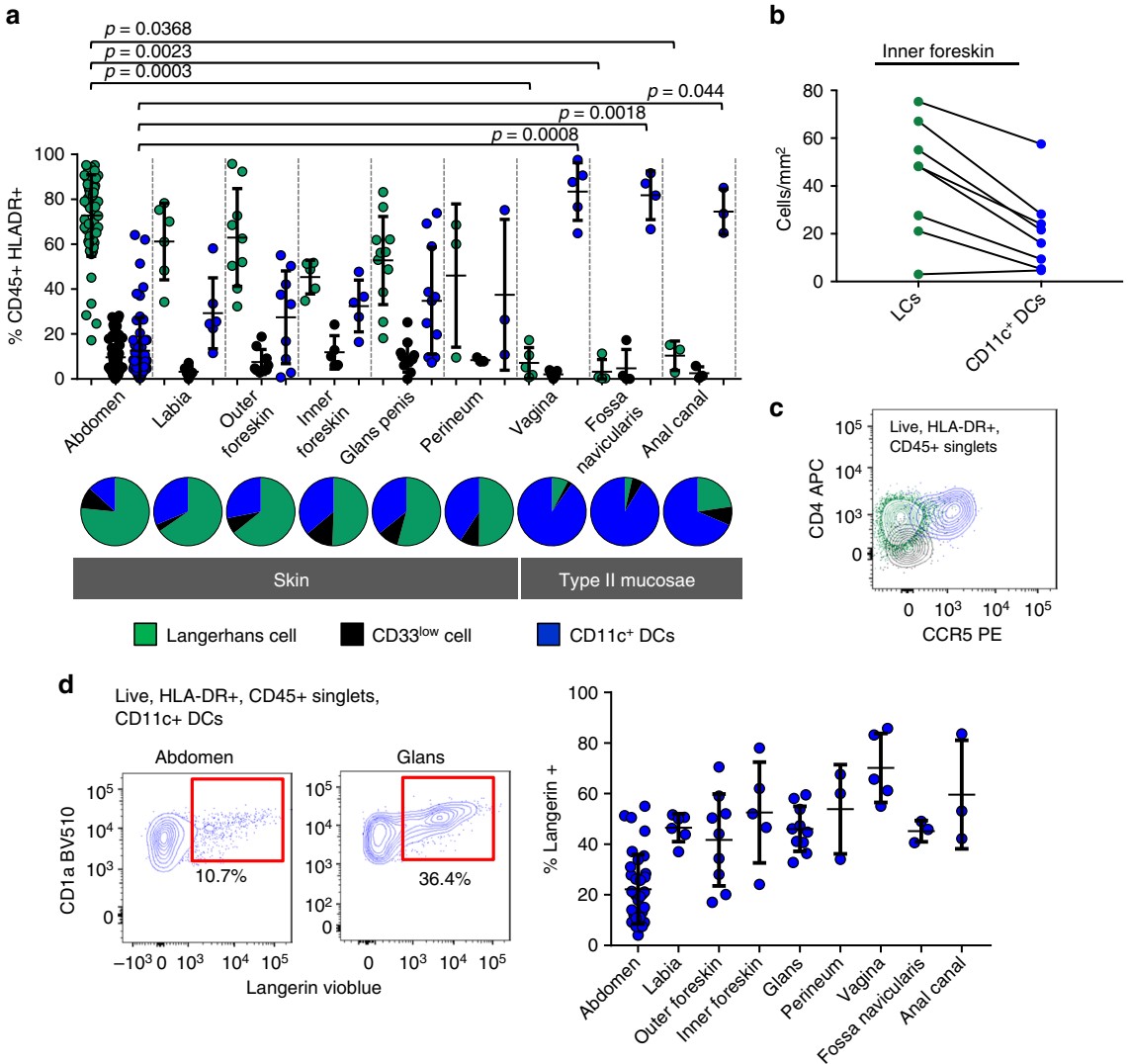

**Fig. 5** Phenotypic analysis of epidermal mononuclear phagocytes in anogenital tissues. **a** Mononuclear phagocytes were liberated from anogenital epidermis and using Type IV collagenase and the relative proportion of each subset within the CD45+HLA-DR+ gate was determined and is shown as a mean ± the standard deviation with each dot representing an individual donor. Statistics for subsets in each tissue were generated using the Kruskal–Wallis test (Abdominal skin = 75, Labia = 6, Outer Foreskin = 9, Inner Foreskin = 5, Glans Penis = 11, Perineum = 3, Vagina = 5, Fossa Navicularis = 4, Anal Canal = 3). A pie chart summarising the average proportions of each epidermal subset in each anogenital tissue is shown underneath. **b** Using fluorescent microscopy we outlined the epidermis of inner foreskin (using pan-keratin stain as a reference) and calculated the density of CD11c+ DCs and LCs within this compartment (n = 7). **c** Representative flow cytometry data using inner foreskin showing that CCR5 is expressed more highly by epidermal CD11c+ DC than LCs or epidermal CD33[low] cells. Representative of n = 5 donors. **d** The percentage of epidermal CD11c+ DCs that express langerin in abdominal and anogenital tissues is shown. Left: Representative data from abdomen and glans penis. Right: data for each tissue is shown as a mean ± the standard deviation with each dot representing an individual donor. Donor numbers for each tissue are as described in part A, except for vaginal tissue and fossa navicularis where five and three donors were analysed, respectively

epithelial surface and take up the virus preferentially to LCs. Furthermore they are most efficient at HIV uptake, support the highest levels of infection and are the most efficient at transferring HIV to CD4 T cells, after de novo infection.

LCs have previously been considered to be the only MNP found within healthy human epidermis. However, in almost all published reports trypsin is used to liberate these cells from tissue and we recently demonstrated that trypsin cleaves several key markers used to discriminate MNPs such as CD11c and CD1c and many pathogen binding receptors such as the key HIV binding receptor CD4[9,14]. We therefore digested healthy human abdominal epidermis with collagenase and identified CD11c+ DCs. Unlike LC, these cells did not express langerin in most cases and they also expressed a greater array of pathogen binding lectin

receptors. Although these cells were less abundant than LCs they were present in easily identifiable proportions in almost every donor we examined (n = 75). We found that epidermal CD11c+ DCs were almost identical to dermal cDC2 by transcriptional profiling and showed similar cell surface marker expression[17]. Furthermore, they morphologically resembled dermal cDC2 and, in abdominal tissue, these cells were usually found deep in the epidermis close to but above the basement membrane i.e. they were not dermal contaminates. In inflamed epidermis CD11c+ IDECs have been described which can be further discerned from LCs by their expression of MR[11–14]. We confirm here that our epidermal CD11c+ DCs also express MR but at much lower levels than dermal cDC2. We therefore hypothesise that the epidermal CD11c+ DCs we observe here (and probably IDECs) are cDC2

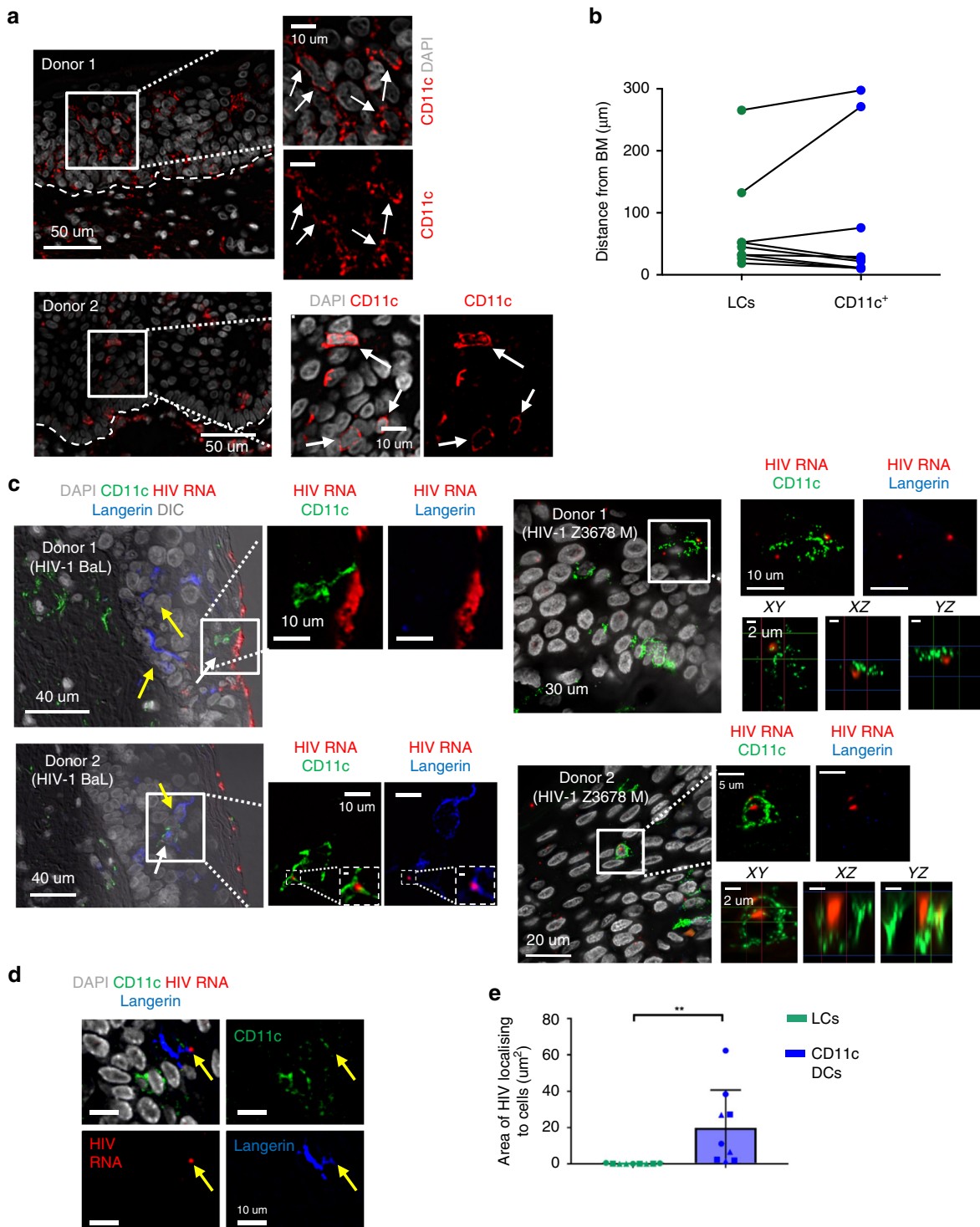

**Fig. 6** Visualisation of anogenital CD11c+ dendritic cells in situ and their interactions with HIV. Human foreskin explants were treated with lab-adapted HIV$_{BaL}$ or transmitted/founder HIV$_{Z3678M}$ or PBS (mock) for 2–3 h before being fixed and paraffin embedded. The tissue was then sectioned and probed for HIV RNA using RNAscope coupled with immunofluorescent staining for CD11c and langerin. Epidermal CD11c+ cells are shown with white arrows and LCs with yellow arrows. Representative images of 5 (HIV$_{BaL}$) or 3 (HIV$_{Z3678M}$) independent donors are shown (see Supplemental Fig. 3a for images from additional donors). **a** In healthy inner foreskin CD11c+ cells were visualised throughout the epithelial layer. The broken line indicates the basement membrane. **b** In 10 donors we manually outlined LCs and epidermal CD11c+DCs and calculated their average distance from the basement membrane using the 'distance map' plug in ImageJ. **c** CD11c+ DCs were observed interacting with HIV$_{BaL}$ and HIV$_{Z3678M}$ (top panels) at the epidermal surface or deeper in the epidermis containing HIV virions (bottom panels). For transmitted founder images Z- stacks were performed with 0.1 micron Z spacing and XY, XZ and YZ projections were generated. Scale bar on the dashed inset image is 1 um. **d** CD11c− langerin+ Langerhans cells could also be detected containing HIV virions. **e** For donors infected with transmitted/founder HIV$_{Z3678M}$, the area of HIV overlap with epidermal CD11c+ cells and LCs was calculated by measuring the area of 'intersection' of each cell type with HIV. 3 sections were measured from three independent donors (nine sections in total). The total area of epidermis measured for each donor was between 23–43 mm$^2$ (54–102 × 20 fields) ($p = 0.0039$) two-tailed Mann–Whitney Test

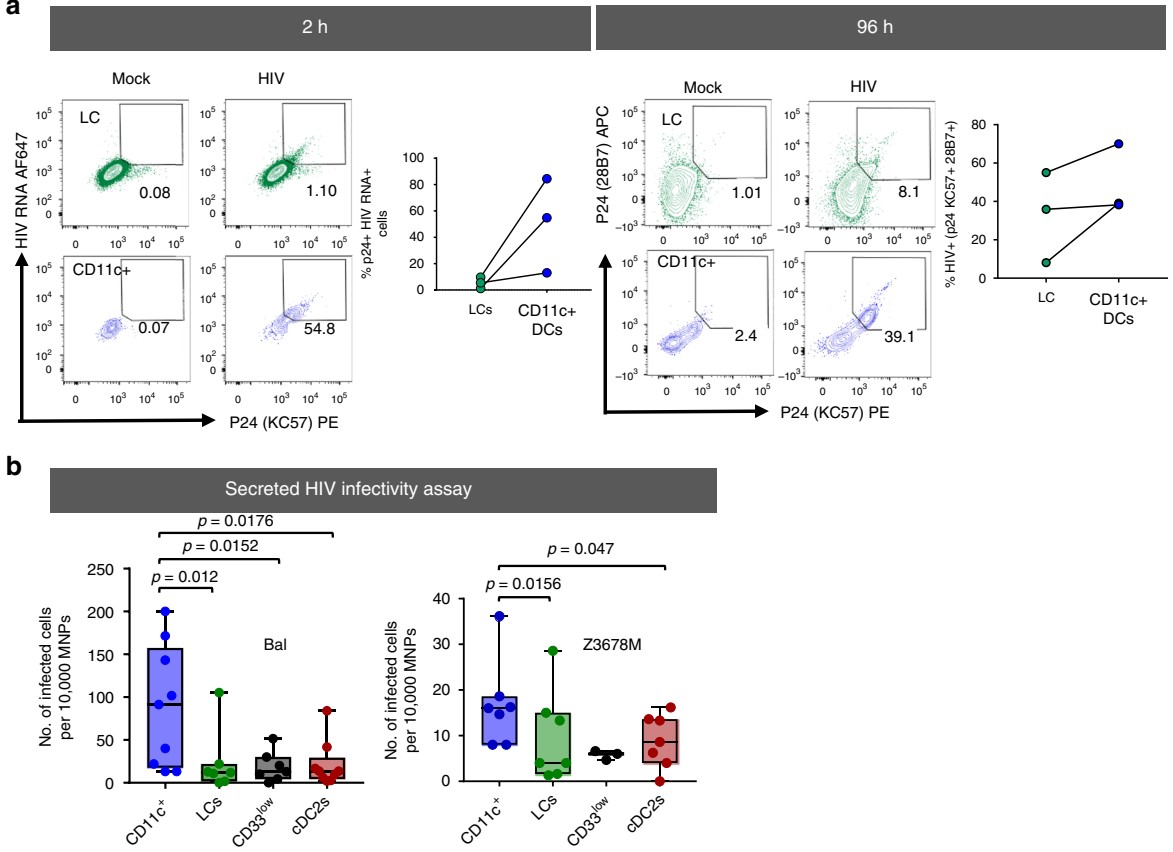

**Fig. 7** HIV infectability and transfer capacity of epidermal mononuclear phagocytes. MNPs were liberated from human epidermis using Type IV collagenase. Cells were either infected as a mixed epidermal population or each subset was FACS sorted and then treated with HIV$_{BaL}$, HIV$_{Z3678M}$, or mock treated for 2 h, after which the virus was thoroughly washed off and cultured in human skin fibroblast conditioned media. **a** Mixed epidermal cells were incubated for 2 h with HIV$_{Z3678M}$, or mock treated then washed 3 times prior to staining with antibodies and HIV p24 and HIV RNA was measured using primeflow. LCs and CD11c+ DCs were gated on and the %HIV p24+ RNA+ cells was analysed. Each dot represents an individual donor with lines indicating the same donor for each subset. Cells were cultured for 96 h and stained for flow cytometry with two antibody clones to HIV p24+ (KC57 and 28B7). LCs and CD11c+ DCs were gated on and the %HIV p24+ cells was analysed. Each dot represents an individual donor with lines indicating the same donor for each subset. **b** Cell supernatants we collected and assayed for HIV infectivity using a TZMBL1 infection assay. The number of infected TZMBL1 cells per 10$^4$ MNPs was calculated and plotted as a box and whisker plots plot. representing, the upper and lower quartile, the central bar represents the median, while the whiskers show the minimum and maximum for each sample, with each dot representing an individual donor. Top panel HIV$_{BaL}$, (CD11c+ = 9, LCs = 7, CD33$^{low}$ = 7, CDC2 = 9). Statistics comparing the amount of HIV produced by each subset were generated using a Mann–Whitney test. Bottom panel HIV$_{Z3678M}$, (CD11c+ = 67, LCs = 67, CD33$^{low}$ = 23, CDC2 = 27). Statistics comparing the amount of HIV produced by each subset were generated using a Wilcoxon matched-pairs test, *p = 0.0313

that have migrated into the lower epidermis from the dermis. This hypothesis is strengthened by the observation that a small proportion of the population express langerin, similar to dermal cDC2[24]. However it should be noted that the two cell types were not identical as, in contrast to epidermal CD11c+ DCs, cDC2 expressed higher levels of CLEC8A, Siglec-1 and lower levels of Siglec-16 on their surface, secreted IL-1$_\alpha$ and significantly lower levels of IL-6, IL-8 and IL-10, and were weaker inducers of T cell proliferation.

LCs have been shown to bind HIV via their CLR langerin and can also become productively infected due their expression of the HIV entry receptors CD4 and CCR5[9,25]. However the role these cells play in sexual transmission is unclear with some groups arguing they act as a natural barrier to HIV infection by rapidly degrading the virus in Birbeck granules[8,10] and others arguing that they directly transfer the virus to CD4 T cells[2–4,9,26]. In order to overcome the poor specificity of HIV antibodies and difficulties in distinguishing fluorescently labelled HIV from autofluorescent structures, we employed recently developed RNAscope technology. This uses florescent nucleotide probes that bind the HIV

genome followed by signal amplification and detected the virus with single virion sensitivity[20]. This technique enabled us to show that epidermal CD11c+ DCs may play a key role in transmission of HIV. Unlike abdominal epidermis, in human inner foreskin we observed these cells throughout the epithelial layer, as opposed to being situated close to the basement membrane. Critically, within 2 h of topical application of purified lab-adapted or transmitted/ founder HIV strains we observed CD11c+ DCs at the epithelial surface appearing to extend processes superficially to capture the virus and CD11c+ HIV containing cells throughout the epithelial layer. Furthermore, more HIV RNA was localised to these cells than LCs in situ. Correspondingly, they also express higher levels of the HIV entry receptor CCR5 as well as the T cell adhesion molecule CD54, the costimulatory molecule CD80 than LCs. They were also much more efficient at HIV uptake than LCs, supported higher levels of productive infection and transferred the virus to CD4 T cells with much greater efficiency. We have previously shown that both monocyte-derived DCs[21,23] and LCs[9] are able to capture HIV and then transfer the virus to CD4 T cells in two successive phases. The first phase of transfer is not

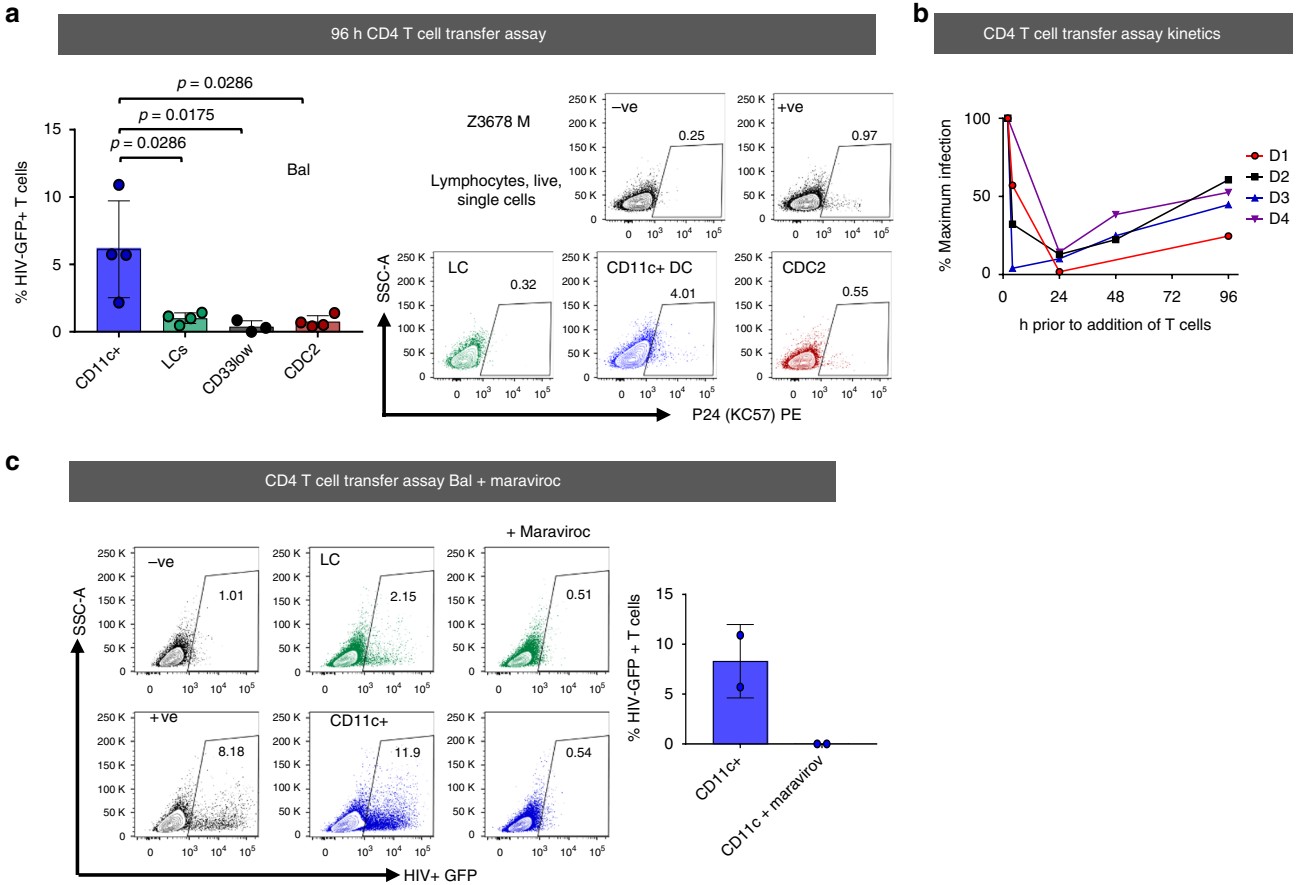

**Fig. 8** HIV transfer capacity of epidermal mononuclear phagocytes. Epidermal MNPs were either infected as sorted subsets and treated with HIV$_{BaL}$, HIV$_{Z3678M}$, or mock treated for 2 h, after which the virus was thoroughly washed off and cultured in human skin fibroblast conditioned media. **a** JLTR cells were added to MNP subsets infected with HIV$_{BaL}$, for 96 h at a ratio of 4 JLTR: 1 DC and co-cultured for a further 96 and the number of infected T cells was determined by flow cytometry. Transfer of HIV$_{BaL}$, to T cells was assessed. The % of GFP+ JLTR cells cultured with each epidermal MNP subset was plotted as the mean ± standard deviation, with each dot representing an individual donor. (CD11c$^+$ = 4, LCs = 4, CD33$^{low}$ = 3, CDC2 = 4). Transfer of HIV$_{Z3678M}$, to T cells was assessed. Primary activated CD4 T cells derived from PBMCs were added and to MNP subsets infected with HIV$_{Z3678M}$, for 96 h at a ratio of 2 T cells:1 DC and co-cultured for a further 72 h and the number of infected T cells was determined by staining for live/dead and HIV P24+ and analysing by flow cytometry. Representative flow plots of p24+ primary T cells after transfer are shown. **b** JLTR cells were added to sorted CD11c+ DCs infected with HIV$_{BaL}$, for 2, 4, 24, 48 and 72 h and co-cultured for a further 96 h and the number of infected T cells was determined by flow cytometry. Infection at 2 h was set at the maximum infection for each donor and time points plotted as a % of maximum infection. **c** Sorted MNP subsets were pre-treated for 1 h with 10 μM maraviroc. After a 2 h infection, cells were washed three times and incubated in the presence of maraviroc. At 96 h, maraviroc was washed off and JLTRs were added then co-cultured for a further 96 h and the number of infected T cells was determined by flow cytometry. Representative plots are shown

dependent of productive infection, but rather on CLR mediated endocytic uptake, and occurs within 2 h and then declines rapidly with time until 24 h post exposure when no transfer to CD4 T cells is observed. The second phase of transfer occurs as a result of CD4/CCR5 mediated productive infection and begins to occur within 72 h of infection and increases with time as newly formed virions bud off from the plasma membrane. Therefore, in order to shown that the more efficient capacity of our epidermal CD11c$^+$ DCs to transfer HIV to CD4 T cells compared to LCs was as result of their increased capacity to become productively infected (correlating with higher CCR5 expression levels), and not as a result of virus being stuck to the cells surface, we performed our 2 phase transfer assay. As expected, similar to monocyte-derived DCs and LC, transfer of the virus from epidermal CD11c$^+$ DCs declined rapidly with time over 24 h before a second wave of transfer was observed from 72 to 96 h. To further confirm that the enhanced transfer capacity of epidermal CD11c$^+$ DCs was due to their higher CCR5 expression we showed that the CCR5 blocking drug maraviroc blocked the ability of these cells to mediate

transfer. Future studies are required to characterise the receptors involved in first phase transfer, likely to be CLRs as with MDDCs (DC-SIGN) and LCs (langerin).

In similar studies to ours, vagina and foreskin LCs have been shown to behave similarly. Using human vaginal explants Hladik and colleagues showed that LCs take up HIV and form clusters with T cells and transfer the virus[3]. Similarly, using human foreskin explants, Ganor and colleagues showed that LCs redistribute within 1 h of HIV exposure, capture the virus and then transfer it CD4 T cells[2]. In addition Patterson and colleagues showed that HIV infection of foreskin explants after topical application was predominantly in CD4 T cells and LCs at the base of the epithelial layer[4]. However, these studies identified LCs using only CD1a and/or langerin. Our data indicates that this is not sufficient to discriminate LCs from epidermal CD11c$^+$ DCs in anogenital epithelium where up to 80% express langerin. Furthermore, we find that CD11c$^+$ DCs are significantly enriched in the epidermis of anogenital tissues. It is therefore probable that these previous studies were observing a mixed population of

epidermal CD11c[+] DCs and LCs. This becomes especially apparent as we observe epidermal CD11c[+] DCs in foreskin capturing HIV at the surface in an identical manner to the CD1a[+]/langerin[+] cells observed by Ganor and colleagues[2]. In our hands, both epidermal CD11c[+] DCs and LCs contained HIV virions in situ and more HIV is localised to epidermal CD11c[+] DCs than LCs. Finally Pena-Cruz and colleagues[27] have recently reported a similar type of HIV binding epidermal DC in vaginal epithelium. However they did not characterise these cells with same phenotypic and functional scrutiny (e.g. CD11c expression) as we report here and further characterisation is required to determine whether it is the vaginal equivalent of the CD11c[+] DCs we describe here throughout the anogenital epithelium.

We also identified a third population of CD45[lo] HLA-DR[lo] cells in the epidermis which, with the exception of their low expressed of CD33 (Siglec-3), were phenotypically very similar to LCs in their surface marker expression. We found that they were transcriptionally very similar to LCs however they were smaller, possessed fewer dendrites, did not secrete cytokines and were very weak inducers of T cell proliferation. Their precise relationship to LCs is unclear.

LCs are known to self-replenish locally within skin[28] and repopulate after inflammation in two waves, the first wave consist of a transient population that are derived from circulating monocytes and the second establish long-term LCs derived from precursors of an unknown origin[29]. Whether the CD33[low] cells we describe here represent a local or recruited precursor LC population requires further exploration in a future study.

In conclusion we have defined an epidermal HIV target cell that may play a critical role in sexual transmission of HIV. The role in viral transmission or immune clearance in epithelial tissues may extend to pathogens other than HIV and requires further investigation. In the context of viral transmission, LCs have been shown to relay HSV to dermal MNPs[30] and it will now be important to examine to role of epidermal CD11c[+] DCs in this process for both HSV and HIV. The location of CD11c[+] DCs in the epithelium also provides the opportunity for early intervention to inhibit transmission.

## Methods

**Sources of tissues and ethical approval.** This study was approved by the Western Sydney Local Area Health District (WSLHD) Human Research Ethics Committee (HREC); reference number (4192) AU RED HREC/15 WMEAD/11. Healthy human abdominal and genital tissues were obtained within 30 min of surgery from a range of patients undergoing plastic surgery (abdomen, labia, vagina), circumcision (foreskin), gender reassignment (glans penis, fossa navicularis) or colorectal surgery (anal canal, perineum) with written consent from all donors.

**Preparation of high-titre purified HIV-1 virus stocks.** Titres of $10^8$–$10^9$ 50% tissue culture infective doses (TCID$_{50}$)/mL of either the laboratory adapted HIV$_{BaL}$ strain, or the HIV$_{Z3678M}$ transmitted founder strain were produced by transfection[22,26,31–34]. 16 million HEK 293 T cells were seeded per T150 flask (Becton Dickinson,Franklin Lakes,New Jersey, USA). The next day, cells were transfected with 80 µg of plasmid DNA pWT/$_{BaL}$ (NIH AIDS Research and Reference Reagent Program, contributed by Dr. Bryan R. Cullen) or pHIV$_{Z3678MTF}$ (Eric Hunter, Genbank accession number: KR820393), and the following all from Sigma-Aldrich, 128 uL 2 M CaCl$_2$, 1 mL (1 mMTris,0.1 mM EDTA, pH8.0), 1 mL Hepes-buffered saline (280 mM NaCl, 50 mM HEPES, pH7.1), 10 µL 0.15 M Na$_2$HPO$_4$ (pH 7.1) and 15 mLDulbecco Modified Eagle Medium (DMEM, Lonza) with 10% Foetal Calf Serum (FCS, Lonza). The next day, media was replaced with DMEM with 10% FCS. After 2 days, media was collected into 50 mL falcon tubes, centrifuged at 1600 g for 20 min and supernatant was collected. Supernatant was concentrated to 1 mL by centrifuging in 300,000 molecular weight cut-off filters (Vivaspin 20, Sartorius, Göttingen, Germany) at 3000 g. Purified high titre HIV-1$_{BaL}$ stocks in the order of $2 \times 10^9$ TCID$_{50}$/mL were generated by infection of SUPT1.CCR5-CL.30 cells (Human Non-Hodgkin's T lymphocyte Lymphoma, contributed by Prof. James Hoxie at the University of PA). CD45[+] MVs were depleted from supernatant using CD45 magnetic beads (Miltenyi Biotech). Virus (18 mL) was incubated at room temperature with 2 mL microbeads for 2 h before adding to the top of a LS column. CD45 depleted virus that flowed through the

column as well as non-depleted supernatants for HIV$_{Z3678M}$ were concentrated further by ultracentrifugation with 1 mL under-layed 20% sucrose cushion and centrifuged at $100,000 \times g$ (Beckman Optima XL-100 K Ultracentrifuge with 70Ti rotor) at 4 °C for 90 min. The 50% tissue culture infectious dose (TCID$_{50}$) values were generated in TZM-BL cells (NIH AIDS Research and Reference Reagent Program, contributed by John Kappes and Xiaoyun Wu) measured by LTR β-galactosidase reporter gene expression after a single round of infection. Endotoxin levels were below the detection limit (*Limulus* amebocyte lysate assay; Sigma) and were negative for tumor necrosis factor alpha (TNF-α), IFN-α, and IFN-β by enzyme-linked immunosorbent assay (ELISA).

**Tissue processing.** MNP were isolated from abdominal tissue using our optimised collagenase based digestion process[15]. Skin was collected immediately after surgery skin was stretched out and sectioned using a skin graft knife (Swann-Morton, Sheffield, United Kingdom) and the resulting skin grafts passed through a skin graft mesher (Zimmer Bionet, Warsaw, IN, USA). The meshed skin was placed in RPMI1640 (Lonza) with 0.14 U/ml dispase (neutral protease, Worthigton Industries, Columbus,OH, USA) and 50 µg/mL Gentamicin (Sigma-Aldrich, St Louis, MO, USA) and rotated at 4 °C overnight. The skin was then washed in PBS and split into dermis and epidermis using fine forceps. Dermal tissue was cut into 1–2 mm pieces using a scalpel. Dermal and epidermal tissue was then incubated separately in media containing 100 U/ml DNase I (Worthington Industries) and 200 U/ml collagenase Type IV (Worthington) at 37 °C for 120 min in a rotator. The cells were then separated from undigested dermal and epidermal tissue using a tea strainer. The supernatants were then passed through a 100 µm cell strainer (Greiner Bio-One, Monroe, NC, USA) and the cells pelleted. The cell pellet was then passed again through a 100 µm cell strainer, and incubated in MACS wash (PBS (Lonza) with 1% Human AB serum (Invitrogen) and 2 mM EDTA (Sigma-Aldrich) supplemented with 50 U/mL DNase for 15 min at 37 °C. The epidermal suspension was spun on a Ficoll-Paque PLUS (GE Healthcare Life Sciences, Little Chalfont, United Kingdom) gradient and the immune cells harvested. Dermal cells were enriched for CD45-expressing cells using CD45 magnetic bead separation (Miltenyi Biotec,San Diego, CA, USA). Cell suspensions were then counted and/or labelled for flow cytometric phenotyping of surface expression markers or for flow sorting.

**Flow cytometry and sorting.** Cells were labelled in aliquots of $2.5 \times 10^6$ cells per 50 µl of buffer, according to standard protocols. Nonviable cells were excluded by staining with Live/Dead™ Fixable Near-IR dead cell stain kit (Life Technologies). Flow cytometry was performed on Becton Dickenson (BD) LSRFortessa flow cytometer and data analysed by FlowJo (Treestar). FACS was performed on a BD INFLUX (140 µm nozzle). Sorted cells were collected into FACS tubes containing RPMI with 10% Fetal bovine serum (FBS). The antibodies were purchased from BD, Miltenyi, BioLegend, Beckman Coulter, eBioscience and R&D Systems as follows; BD: CD45 BV786

(HI30), HLA-DR, BUV395 (G46-6), CD1a BV510 (HI149), CD33 APC (WM53), CXCR4 PE (12G5), CD4 APC (RPA-T4), CD80 PE (L307.4), CD83 APC (HB15e), CD86 APC (2331

(FUN-1), CCR5 PE (2D7), DC-SIGN APC (DCN46), Clec12A AF647 (50C1), EpCAM APC (EBA-1), MR APC & BV510 (19.2), Mouse IgG1 APC, Mouse IgG1 PE. Miltenyi: Clec7A PE (REA515), Clec9A PE (8F9), CD1c PE-Vio770 (AD5-8E7), Siglec-5 APC (1A5), Langerin

Vioblue (MB22-9F5). Bio Legend: Siglec-1 PE (7-239), DEC205 PE (HD30). Beckman Coulter: Langerin PE (DCGM4), MR PE (3.29B1.10). eBioscience: CD91 eFluor660 (A2MRa2). R&D Systems: L-SIGN PE (120604), DCIR PE (216110), Clec4D PE (413512), Clec4G APC (845404), Clec5A APC (283834), Clec5C APC (239127), Clec6A APC (545943),

Clec10A PE (744812), Clec14A APC (743940), Siglec-6 APC (767329), Siglec-9 APC (191240), Siglec-16 APC (706022). After Live/Dead Near- IR and surface staining, HIV was detected by flow cytometry by incubating cytofix/cytoperm solution (BD) for 15 min, washing with permwash (BD) and incubating with antibodies to P24 PE (KC57, Beckman Coulter) and dual staining with either P24 APC (28b7, Medimabs) or utilising PrimeFlow™ (Thermofisher) for increased sensitivity. PrimeFlow™ (AF647) was performed as per the manufacturer's instructions with a few minor changes. The target probe incubation was increased to 3 h, the pre-amplification, amplification and label probe were increased to 2 h. The concentration of the target probe was also increased from 1/20 to 1/10.

**Cytometric bead arrays.** Culture supernatants were collected from sorted epidermal MNP subsets after 24 h culture with TLR agonist treatment (Pam2Cys (5 µM), R848 (1 µg/ml) and LPS (1 µg/ml)), and stored at −80 °C until use. The concentrations of TNF, IP-10, IL-10, IL-8, IL-6, IL-1$_\beta$ and IL-1$_\alpha$ were analysed by CBA (BD Biosciences), according to the manufacturer's instructions for the Human Soluble Protein Master Buffer Kit. Sample data was acquired with the BD FACSCanto II flow cytometer and analysed using FCAP array software (BD Biosciences). The standard range of detection for each cytokine was between 10–2500 pg/mL.

**RNA Seq**. Cells isolated by flow cytometry ($n = 200$ cells per population) were captured directly into lysis buffer containing 0.1% triton X-100, 1U RNaseOUT RNase inhibitor (Life TechnologiesThermoFisher Scientific, Mulgrave, Australia), 2.5 mM each dNTP mix, 2.5 uM anchored oligo-dT primer (5′-AAGCAGTGG-TATCAACGCAGAGTACT30VN-3′, Sigma-Aldrich, Castle Hill, Australia) in a total volume of 20 μl. Full-length mRNA was reverse transcribed and amplified using the Smart-seq2 protocol of Picelli et al.[35,36] Aliquots of each cell lysis mix were first heated to 72 °C for 3 min then placed on ice immediately. A first strand reaction mix (6 μl) containing Superscript II reverse transcriptase (100 U, Life Technologies-ThermoFisher Scientific), TSO primer (5′-AAGCAGTGGTAT-CAACGCAGAGTACATrGrG+G-3′, 1 uM, Exiqon, Vedbaek, Denmark), 1 M betaine (Sigma-Aldrich), 6 mM MgCl2 (Sigma-Aldrich), 5 mM DTT (Life Technologies-ThermoFisher Scientific), RNaseOUT (10 U, Life Technologies-ThermoFisher Scientific) and 2 μl 5X first strand buffer (Life Technologies-ThermoFisher Scientific), was added to each sample. Transcripts were reverse transcribed by incubating at 42 °C for 60 min, followed by 10 cycles of 50 °C, 2 min and 42 °C, 2 min. The resulting cDNAs were subsequently amplified by adding 12.5 μl KAPA HiFi 2X HotStart ReadyMix (KAPA Biosciences, Wilmington, MA), 10 μM ISPCR primer (0.25 μl, 5′AAGCAGTGGTATCAACGCAGAGT-3′, Sigma-Aldrich) and 2.25 μl H2O and heating to 98 °C, 3 min, followed by 20 cycles of 98 °C, 20 s, 67 °C, 15 s, 72 °C, 6 min. After a final incubation at 72 °C, 5 min, samples were held at 4 °C until purification. Samples were purified by addition of Agencourt AMPure XP beads (1:1 v/v, Beckman Coulter, Lane Cove, Australia), 8 min, room temperature, followed by magnetic bead capture and two washes with 200 μl 80% ethanol. Beads were dried and cDNAs were eluted into 17.5 μl elution buffer EB (Qiagen, Chadstone, Australia). Products were checked on Agilent Bioanalyser High Sensitivity DNA chips and were estimated by Quant-iT PicoGreen dsDNA Assay (Life TechnologiesThermoFisher Scientific). Sequencing libraries were prepared from 1 ng cDNA product using Nextera XT reagents (Illumina, Scoresby, Australia) in accordance with the supplier's protocol except that PCR was limited to eight cycles to minimise read duplicates, and final library elution was performed using 17.5 μl elution buffer EB. Libraries were pooled and sequenced on the Illumina HiSeq 2500 platform. Reads were aligned using STAR v2.5.3[37] to the GRCh38 genome and were summarised to gene counts as annotated by GENCODE v26[38]. Gene counts were normalised using TMM[39] and variance stabilised using DESeq2[40]. Barnes-Hut tDistributed Stochastic Neighbor Embedding with perplexity 10 was used to perform dimension reduction on the 231 genes with average count greater than 20 and stabilised variance 50% higher than the average.

**QPCR**. Total unamplified RNA was DNase I treated (Promega, Madison, WI) and reverse transcribed using oligod(T) and superscript III (Invitrogen). The cDNA was subject to QPCR using primers for CTSD, CD1E, NLRP3, ILR1R2, RNASE1, CADM1, CLDN1, MMP12, ERG and GAPDH (Sigma-Aldrich) and SYBR Green (Invitrogen). The data was analysed using the a standard curve method[17,21,22,26,31-34,41].

**T cell proliferation assays**. Epidermal MNP subsets were FACS sorted and added to allogeneic T cells isolated from PBMC using CD3+ microbeads (Miltenyi) then stained with Celltrace™ Violet (Thermofisher) as per the manufacturers instruction at a ratio of 1:10 and cultured for 7 days. Positive (LCs + PHA (5 μg /mL, Sigma-Aldrich) and negative controls were also made up. Cells were then stained for Live/Dead NIR and CD3 FITC, CD4 PE and CD8 APC and analysed on the BD FACSCanto II.

**Immunofluorescent staining of tissue**. Four micrometre paraffin sections were baked at 60 °C for 40 min, dewaxed in xylene and 100% ethanol and air dried. Between all steps described herein, sections were washed three times in TBS (Amresco, Cat: 0788) filled jars placed on a rotator for a total of 10 min. Antigen retrieval was then performed using a ph9 antigen retrieval buffer (DAKO) in a decloaking chamber (Biocare) for 20 min at 95 °C. Sections were incubated with 3% H2O2 for 10 min and then blocked for 30 min (0.1% Saponin, 1% BSA, 10% donkey serum). Primary antibody incubation was at 4 °C overnight. Antibodies for primary detection include: Abcam – rabbit CD11c (EP1347Y), rabbit MR (polyclonal), rabbit HLA-DR (EPR6148), mouse CD1c (2F4), mouse CollagenVII (LH7.2); R&D - goat langerin (AF2088). Sections were then incubated with secondary antibodies for 30 min at RT. Donkey secondary antibodies (molecular probes) against rabbit, mouse or goat were used and were conjugated to either alexa fluor 488, 546 or 647 or HRP. For HRP-labelled secondaries tyramide-FITC (perkin elmer) was then applied for 10 min at RT. Sectioned were then DAPI stained and mounted using slowfade diamond antifade (molecular probes).

**Immunofluorescence microscopy**. Tissue sections were imaged using x40 objective on either a Deltavision restoration microscope with a Photometrics CoolSnap QE camera (GE Healthcare, IL) (images in Fig. 2), or the InCell 2200 (GE Healthcare, IL) (images in Fig. 5). Images were then deconvolved and stitched using either SoftWoRx (version 6.1.3) for Deltavision images, or InCell analyser software for InCell images (GE Healthcare, IL). All subsequent image analysis was performed using Fiji (Madison version).

In some cases multiple Z-planes were imaged using a x60 objective on the Deltavision System with 0.1 um z-spacing (Transmitted/Founder HIV images in Fig. 6e and Supplementary Fig. 3). Huygens Professional 18.10 (Scientific Volume Imaging, The Netherlands, http://svi.nl) CMLE algorithm, with SNR:20 and 40 iterations, was used for deconvolution of Z-stacks. XY, XZ, YZ projections were generated using the Huygens Orthoslicer feature.

**Ex vivo inner foreskin explants**. Healthy Inner foreskin explants were infected with either HIVBal or Transmitted/Founder HIV1 Z3678M using an explant setup. Briefly, pyrex cloning cylinders (8 × 8 mm, Sigma-Aldrich) were dipped in histoacryl surgical glue (B Braun) and placed over the mucosal surface of the tissue. We then cut around the cloning cylinders and loaded the explant tissues onto gelfoam sponges (Pfizer) soaked in culture media in a 24-well plate. Culture media consisted of 10 μM HEPES (Gibco), non-essential amino acids (Gibco), 1 mM sodium pyruvate (Gibco) 50 μM 2-Mercaptoethanol (Gibco), 10 μg/ml Gentamycin (Gibco) and 10% fetal calf-serum, all diluted in RPMI (Lonza). Wells were filled with culture media to just below the tissue surface level and PBS was then added to the inner chamber of the cloning cylinders to prevent the surface from drying and to assist with setting of the glue. For infection, PBS was removed from the chambers and virus at a TCID50 of 3500 (diluted in 100 μl PBS) was added for 2–3 h at 37 °C. At the end of the culture period, solutions were aspirated from the cloning cylinder and the surface was washed three times with PBS to remove virus resting on the tissue surface. Cloning cylinders were removed and placed in 4% PFA (diluted in PBS) (electron microscopy sciences) for 18–24 h at room temperature. Prior to paraffin embedding, tissues were trimmed along the mark left by the cloning cylinder so that only the infected area remained.

**RNAscope**. Detection of HIV RNA was performed using the 'RNAscope 2.5HD Reagent Kit-RED' (Cat: 322360, ACD Bio) as described previously[21] with custom probes (consisting of 85 zz pairs) against HIV-1BaL (REF: 486631, ACD Bio) spanning base pairs 1144–8431 of HIV-1BaL sequence. This probe was also used for the detection of HIV-1 Z3678M. In the protocol described below, reagents from 'RNAscope 2.5HD Reagent Kit-RED' are indicated with 'RNAscope kit' in brackets after the reagent name. Unless otherwise indicated wash steps with each solution were carried out for 2 min on a rotator set to low. All 40 °C incubations were carried out in a hybridisation oven (HybEZ Hybridization System (220VAC) With ACD EZBatch Slide System, ACD Bio).

Four micrometre paraffin sections from infected tissue blocks were baked, dewaxed and antigen retrieval performed as described above in 'Immunofluorescent Staining of Tissue'. Sections were then washed with Milli-Q water, followed by TBS (Amresco, Cat: 0788) and then immersed in a 300 mM Glycine (Amresco, Cat: 0167) solution, (diluted in TBS) for 1.5 h at room temp. Sections were washed in TBS, followed by Milli-Q water, then dipped 3–5 times in 100% ethanol, allowed to air dry then encircled with a hydrophobic pen. Sections were then covered with protease pre-treatment 3 (RNAscope kit) and incubated for 30 min in a hybridisation oven set to 40 °C. Sections were washed twice in Milli-Q water and a custom probe against HIV-1BaL (REF: 486631, ACD Bio) or DapB negative probe control (cat: 310043, RNAscope kit,) was applied to sections for 2 h at 40 °C. Sections were washed twice with RNAscope wash buffer (RNAscope kit). Six amplification reagents labelled Amp 1–6 (RNAscope kit) were then applied sequentially to the tissue sections with two washes in RNAscope wash buffer in between each incubation. Amp 1 = 30 min at 40 °C; Amp 2 = 15 min at 40 °C; Amp 3 = 30 min at 40 °C; Amp 4 = 15 min at 40 °C; Amp 5 = 30 min at room temp; Amp 6 = 15 min at room temp. HIV RNA was then developed using a Fast Red chromagen solution made by mixing Red-B and Red-A (RNAscope kit) in a 1:60 ratio and applying to sections at room temp for 2 min. Sections were then washed in Milli-Q water followed by 0.1% Tween-20 (diluted in TBS).

Immunofluorescent staining against cell surface markers was then carried out as described in 'Immunofluorescent Staining of Tissue', starting from the application of 3% H2O2 to tissue sections. Sections were then imaged on the InCell 2200 using channels FITC (Ex: 490/20, Em: 525/36), Cy3 (Ex: 543/22, Em: 605/64), Cy5 (Ex: 645/30, Em: 705/72).

**MNP-CD4 T cell HIV transfer assay**. Epidermal mononuclear phagocytes were isolated from abdominal skin using collagenase digestion and LCs, CD11chiDCs and CD33low cells sorted by FACS. A minimum of $3 \times 10^4$ of each cell population was then infected with HIVBaL at MOI = 1 for 2 h and the virus washed off using 3x PBS washes. Infected cells were then cultured human skin fibroblast conditioned media to enhance cell survival[15] and JLTR CD4 T cells (which express GFP under control of the HIV-1 promotor[10] were added at ratio of 4:1 after 96 h and co-cultured for a further 96 h. The percentage of GFP+ JLTR cells were then determined by flow cytometry. Transfer assays with HIVZ3678M were performed as above, except CD4 T cells isolated (CD4 T cell negative selection kit, Stemcell) from PBMCs activated for 3 days with PHA (5 μg/mL) and IL-2 (150 IU/mL, Peprotech) were added at a ratio of 2:1 and co-cultured for a further 72 h prior to staining with Live/Dead Near-IR and intracellularly for P24 (KC57)-PE. For Inhibition experiments, sorted MNPS were pre-treated for 1 h with 10 μM Maraviroc (kindly gifted by Paul Gorry, Melbourne), HIV was added for 2 h, cells were washed and cultured

for a further 96 h in the presence of maraviroc. At 96 h, maraviroc was washed off prior to the addition of JLTR cells for a further 96 h.

**Statistical analysis**. Statistical analysis and graphs were produced using Graphpad Prism (version 7.02) and show mean ± standard deviation unless stated otherwise. Normalised distribution was not assumed to account for donor-to-donor variability, thus statistics were produced using an unpaired nonparametric Mann–Whitney $t$ test or a Wilcoxon test for paired data. Results were considered statistically significant when $P \leq 0.05$. Significance levels were ns; $*p < 0.05$; $**p < 0.01$; $***p < 0.001$; $****p < 0.0001$ as suggested by Graphpad Prism.

**In situ quantification**. In situ quantification was performed using Fiji (Madison version) and included distance measurements and measurements of HIV-cell interactions.

The distance of Epidermal CD11c+ cells and Langerhan's Cells to the basement membrane (Fig. 6b) was calculated as follows: (1) the basement membrane was manually outlined and a Euclidean Distance Map (EDM) map was generated emanating from the basement membrane. (2) Each cell type was manually outlined and stored in the ImageJ Region of Interest (ROI) manager. (3) The mean pixel intensity of each ROI on the EDM map was calculated. (4) Distance measurements were extracted by converting pixel units to microns using the original image resolution.

HIV-cell interactions (Fig. 6e) were calculated as follows: (1) The epidermis was manually outlined and stored in the ROI manager. (2) Binary masks of Epidermal CD11c+ cells (CD11c+), LCs (CD11c-Langerin+) and HIV were generated using a manual threshold. (3) The ImageJ calculator 'AND' function was used to extract the intersection of CD11c+ cells and HIV, or LCs and HIV. (4) The area of the intersection mask was measured within the epithelium to estimate the overlap of each cell type with HIV.

**Reporting summary**. Further information on research design is available in the Nature Research Reporting Summary linked to this article.

## Data availability
The data that support the findings of this study are available from the corresponding author upon reasonable request. Source data used to generate Figs. 1, 4–7 and Supplemental Fig. 2 are provided in the Source Data file. The RNA sequencing data have been deposited in the Gene Expression Omnibus (GEO) database under accession code: GSE130804.

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

## Acknowledgements

Flow cytometry and cell imaging were performed in the Flow Cytometry Core Facility and Cell Imaging Core Facility that are supported by Westmead Institute, Westmead Research Hub, Cancer Institute New South Wales and National Health and Medical Research Council. Immunofluorescence microscopy imaging was also performed in the

ithree Institute, University of Technology Sydney. This work was funded by the National Health and Medical Research Council (Australia) the Wellcome Trust (WT107931/Z/15/Z), the Lister Institute and the NIHR Newcastle Biomedical Research Centre. This project has been funded in part with federal funds from the National Cancer Institute, National Institutes of Health, under Contract No. HHSN261200800001E. The content of this publication does not necessarily reflect the views or policies of the Department of Health and Human Services, nor does mention of trade names, commercial products, or organizations imply endorsement by the U.S. Government.

## Author contributions

K.M.B. guided all the experiments in the manuscript that were not miscopy based. R.A.B. made the initial discovery of the three populations and contributed to flow cytometry experiments. H.B. guided all flourescence microscopy experiments and developed RNAscope. J.W.R. helped with all HIV uptake, infectivity and transfer assays, sample preparation for RNAseq and carried the QPCR validation. H.R. helped with cytokine release assays, T cell proliferation assays, some HIV transfer assays and microscopy distance from basement membrane measurements. Dinny Graham guided the RNAseq. E.P.l conducted the RNAseq analysis. J.F. conducted the Giemsa staining for morphological examination of MNP subsets. T.M.P. helped with MNP visualization in situ by florescence microscopy. N.R.T. helped with cytometric bead array experiments to measure cytokine secretion. C.R. and O.T. optimised transfections and growing of the transmitted-founder virus. C.M.D. gave extensive help with tissue processing and guided cell isolation protocols. N.N. provided input into the infection experiments and MNP-CD4 T cells transfer assays. L.B., M.P.K., A.J.B., M.P.W., P.H., J.L., M.P.G. and G.C. provided human tissue specimens and intellectual input. J.D.E. and M.J.C. helped guide RNA scope experiments. P.U.C. provided intellectual into DC ontogeny. E.H. provided the clinical transmitted founder isolate and intellectual input into the best strain to use. M.A.H. and A.L.C. provided significant intellectual input. A.N.H. conceived of and guided the study.

## Additional information

**Competing interests:** The authors declare no competing interests.

