## [Peer Review File · Nature Communications]

Editorial Note: This manuscript has been previously reviewed at another journal that is not operating a transparent peer review scheme. This document only contains reviewer comments and rebuttal letters for versions considered at Nature Communications .

REVIEWERS' COMMENTS:

Reviewer #1 (Remarks to the Author):

the authors have addressed my concerns

Reviewer #2 (Remarks to the Author):

This is starting as an interesting story aiming to decipher the heterogeneity of the mononuclear phagocyte (MNP) population in the human epidermis. The authors showed that in addition to classical LC population, two others populations of cells either expressing high level of CD11c related to DC2 and another one with low CD33, DR and CD45 related to LCs can be found. Finally, the authors found that the CD11c+ DCs are enriched in the epithelium of anogenital tissues and are able to better capture HIV virions and transfer virus to CD4+ T cells, suggesting that they play a role in HIV infection. Although interesting, the manuscript suffers from few major limitations: First, the low CD33, DR and CD45 population related to LCs is poorly characterized and the part describing it appears as a side story out of scope of the study. Second, while the flow cytometry data describing MNP is convincing and of interest, the microscopy data related to HIV are barely convincing (Figure 6) as well as the data presented in Figure 7. Lastly, Figure 2 data are not analyzed properly and in details. This part should be extended: DEG, clustering, pathway analysis, validation by PCR and others techniques.

Reviewer Rebuttal:

Reviewer #1: Happy with all modification, no rebuttal required

Reviewer #2:

1. Reviewer: 'the low CD33, DR and CD45 population related to LCs is poorly characterized and the part describing it appears as a side story out of scope of the study'.

Response: This manuscript has two components; (i) the definition of previously unidentified MNP subsets in healthy human epidermis (**Figures 1-4**); (ii) the role that one of these subsets (CD11c⁺ epidermal cells) plays in transmission of HIV (**Figures 5-7**). The observation of a 'CD33, DR and CD45 population' clearly relates to the first component of this manuscript. We believe they should be included as we are the first group to describe these cells. It is likely that these cells have been previously missed as they are present in very low proportions in most donors (**Fig 1C**). We were able to identify them as we obtain very large abdominal skin explants (up to the size of 4 A4 sheets of paper). It was when processing these very large explants that we began to notice a small separate cell population in the CD45, HLA-DR gate. This led us to gate them out and investigate them separately. Interestingly, the CD33 low cells are transcriptionally almost identical to LCs (**Fig 2**), however they differ to LCs in their morphology (**Fig 3C**) and are functionally inferior APCs (**Fig 4**). We believe they may be the Langerhans cell precursor that have been predicted to exist but nobody has been able to find them. We therefore feel that this is a significant finding and does fit the context of the first half of this manuscript.

2. Reviewer: 'while the flow cytometry data describing MNP is convincing and of interest, the microscopy data related to HIV are barely convincing (Figure 6) as well as the data presented in Figure 7'.

We do not agree with this statement.

Figure 6: We have now included Z stack images for Figure 6 and Supplementary Figure 4 for several donors which clearly shows that HIV co-localises with epidermal CD11c⁺ cells within 2 hours of topical exposure.

Figure 7: This robustly demonstrates that, compared the LCs, epidermal CD11c⁺ DCs preferentially take up HIV at 2 hours (**Fig 7A, left**) and go on to support higher levels of productive infection using two assays (**Fig 7A, right and 7B**). Critically, these cells transfer HIV to CD4 T cells with greater efficiency than LCs (**Fig 7C**) in two distinct phases (**Fig 7D**). Finally we show that late phase transfer can be blocked using the CCR5 inhibitor drug maraviroc demonstrating that transfer to CD4 T cells is dependent on viral entry and infectivity (**Fig 7D**). All key results were reproducible using a transmitted founder strain. All these experiments were carried out using ex vivo derived human cells and represent a huge body of work by a post doc and PhD student which has 15 months since receiving feedback on the original manuscript submitted to Nature Medicine.

2. Reviewer: Figure 2 data are not analyzed properly and in details. This part should be extended: DEG, clustering, pathway analysis, validation by PCR and others techniques.

We carried out the RNAseq as requested by Reviewers 1 and 2 to answer a specific question as to how the epidermal CD11c⁺ DCs and CD33^{low}MNPs transcriptionally relate to other known human tissue MNPs. Thus, was presented the clustering shown in **Fig 2A** to specifically answer this question. We have now also used moderated linear models to identify genes whose high or low expression characterized each subset. The heat maps display expression values of the top 40 genes that uniquely identify; CD33^{low} MNPs and LCs; cDC2 and epidermal CD11c; cDC1; and monocyte-derived macrophages and tissue resident Macrophages (**Fig 2B**).

We have also chosen some genes identified from the heat maps that discriminate the clusters of cell populations and confirmed that these are differentially expressed in 5 independent donors (**Supplemental Figure 2**). Note that as these are small cell populations derived from human tissue we did not have any of the RNA or cDNA left that was used for the RNAseq for direct qPCR validation on the same donors. Instead we used cells from 5 different donors sorted previously. This adds weight to the findings as it means that these genes are differentially expressed across as many as 9 donors (4 by RNAseq and 5 by QPCR).

Unfortunately our attempts to generate cDNA from the CD33^{low}MNPs failed as these cells are present in such tiny proportions. We could only detect gene expression for 1 gene in 1 donor (CLDN1). Importantly, this did validate the RNAseq data. We had similar problems for cDC1 where we could only detect gene expression in 2 (or sometimes 3 donors). In the case of cDC1 this is a lesser concern as they are not a focus of this manuscript and there are a plethora of transcriptomic cDC1 studies in the literature. For the CD33^{low}MNPs this is more unfortunate. However to re-isolate 4 more donors of CD33^{low}MNPs and then amplify the cDNA will take us up to 10 months. These cells are present in very low proportions in almost all donors (**see Fig 1C**) and we can only obtain enough cells to sort in about 1 in 10 donors. Thus, we would need to process about 40 abdominal tissue samples of which we obtain about 1 per week. We hope very much that in the case of this cell population the RNAseq data will be allowed to stand for itself, not validated by QPCR. If this is unacceptable then, at the reviewer's suggestion, we will need to remove the CD33^{low}MNPs from the manuscript. I believe this would be great shame for the reasons described above.

Importantly, the validation of the epidermal CD11c⁺ DCs was solid and these cells are the central focus of the manuscript.